# Psychometric Properties of the Health Professionals Communication Skills Scale in University Students of Health Sciences

**DOI:** 10.3390/ijerph17207565

**Published:** 2020-10-18

**Authors:** Rocío Juliá-Sanchis, María José Cabañero-Martínez, César Leal-Costa, Manuel Fernández-Alcántara, Silvia Escribano

**Affiliations:** 1Department of Nursing, Faculty of Health Science, University of Alicante, 03690 San Vicente del Raspeig (Alicante), Spain; rjulia@ua.es (R.J.-S.); silvia.escribano@ua.es (S.E.); 2Department of Nursing, Faculty of Health Science, University of Murcia, 30100 Murcia, Spain; cleal@um.es; 3Department of Health Psychology, Faculty of Health Science, University of Alicante, 03690 San Vicente del Raspeig (Alicante), Spain; mfernandeza@ua.es

**Keywords:** health communication, nursing, psychometrics, scale, students, validation studies

## Abstract

Communication is one of the determining factors of healthcare quality; however, a health model that prioritizes clinical over non-technical skills remains prevalent. The aims of this article were: (a) to validate a communication skills scale in a sample of fourth-year nursing degree students from two Spanish universities and (b) determine their perception of communication skills. The study included 289 fourth-year nursing undergraduate students with a mean age of 22.7 (*SD* = 4.87) years; 81.7% were female. The Health Professionals Communication Skills Scale (HP-CSS) questionnaire was adapted for use among nursing students. We analysed the psychometric properties and relationships with the variable attitudes toward communication skills. The HP-CSS showed a high internal consistency (0.88) and good fit of data to the model (*TLI* = 0.98; *CFI* = 0.97; *RMSEA* = 0.05 [95% CI = 0.04–0.06]). The total score and subscale scores correlated with the variable attitude towards communication skills. High scores were obtained for the students’ perception of communication skills. The HP-CSS is a valid and reliable tool to assess the communication skills in nursing students. This scale provides university teachers with a rapid and easily applied instrument to assess the level of communication skills and relationship with patients.

## 1. Introduction

Health care has undergone profound structural and technological transformations since the second half of the 20th century. These include the gradual introduction of the concept of person-centred care [1] giving users a greater role in the decision-making and health care processes [2]. Nevertheless, the care paradigm continues to be dominated by a biomedical model that prioritizes scientific and technical knowledge of disease over the quality of interactions with patients [2,3]. It has been reported that healthcare professionals are becoming more technically competent; while their training in non-technical skills, including interpersonal skills such as team work or communication [4], are seldom taught [5]. A recent study [6] evaluated the communication capacities of nursing students in simulated settings and attributed their low scores, to the predominant focus of their training on clinical skills [5].

Communication is an essential element of daily nursing practice and a key determinant of the quality of their care [7], although this is not always recognized by nurses themselves [8]. The UK Department of Health of England [9] identified communication as one of six essential nursing skills to achieve a high quality of care, and good communication has been associated with improved health outcomes [10], better adherence to treatment [11], and general patient satisfaction [12]. Conversely, negative outcomes (e.g., increased hospital stay) and higher costs have been associated with deficient communication [13]. It therefore appears important to devote more resources and time to the acquisition by future nursing professionals of communication skills using effective methodologies [6,14].

The European Higher Education Area [15] has emphasized the acquisition of practical knowledge, attitudes, and skills [16], implying the need for corresponding changes in educational methods and evaluation systems [17]. However, there is a lack of instruments to assess the communication skills acquired by students. In the literature, measurement tools that assess general social skills are often used to examine the effectiveness of programs that promote communication in health students [18]. Instruments that assess specific communication skills in therapeutic relationships often require external observation [19,20].

The self-administered Health Professionals Communication Skills Scale (HP-CSS) yields information on four dimensions: empathy, informative communication, respect, and assertiveness, and it has demonstrated adequate psychometric properties in health care professionals [21]. To our best knowledge, however, it has not been validated in nursing degree students, although it has been applied in this setting [22]. Before using a scale in a new population, it is advisable to examine its performance and psychometric properties in that group [23]. Therefore, the objective of this study was to validate the HP-CSS in a sample of nursing degree students from two Spanish universities and to determine the perception by students of communication skills and relationships with patients.

## 2. Materials and Methods

### 2.1. Design, Setting and Sample

We conducted an instrumental study, designed for the generation or adaptation of new measurement instruments and the analysis of their psychometric properties [24].

All students in their fourth year of nursing at two Spanish universities were invited to participate. In the Spanish university environment, the fourth year is the last course of the nursing degree studies. Fourth-year students were chosen to ensure that, in spite of possible differences in their curricula, both universities had addressed content related to communication skills in different situations, as well as having carried out more than 75% of their clinical practice. Data were gathered between June and October 2019. Inclusion criteria were: (1) enrolment in the nursing degree, (2) participation in a course in the fourth year of the degree, and (3) fluency in written and spoken Spanish.

Out of 800 students who met the inclusion criteria, 289 completed the questionnaire and finished the study, a response rate of 36.75%: 65 (21.45%) were enrolled in academic year 2018/19 (52 (80%) from the University of Alicante and 13 (20%) from the University of Murcia, and 224 (78.58%) in academic year 2019/20 (206 (91.96%) from the University of Alicante and 18 (8.03%) from the University of Murcia).

### 2.2. Measures

An ad hoc questionnaire was used to gather sociodemographic data on age, gender, nationality (Spanish and others), and marital status (single, married/common-law partner, separated/divorced). Participants were asked if they had previously received training/education on communication during the Nursing Degree or in other contexts.

The Family Affluence Scale (FAS_III) [25] was used to assess family socioeconomic status. This questionnaire evaluates the economic well-being of the family structure through 6 items on: number of cars (0, 1, 2 or more), number of computers (0, 1, 2, 3 or more), whether the participants have their own room (No = 0, Yes = 1), the number of family holidays outside the country in the last 12 months (0, 1, 2, 3 or more), whether they have a dishwasher (No = 0, Yes = 1), and the number of bathrooms in the household (0, 1, 2, 3 or more). The score ranges from 0 to 13, with a higher score indicating a higher socioeconomic level. It has a good construct validity with respect to household income for most validated countries (*Eta*^2^ > 0.30) and a high test–retest reliability (*r* = 0.90).

A Spanish adaptation of the Attitudes towards Medical Communication Scale [26], developed in a Canadian context, was used to evaluate the attitudes of students towards communication in the health setting (unpublished results). It is a one-dimensional 12-item instrument on general attitudes towards communication, with responses on a 5-point Likert-type scale (1 = strongly disagree to 5 = strongly agree). The total score ranges from 12 to 60, with a higher score indicating a more positive attitude towards communication. The original version evidenced adequate internal consistency of 0.74.

The Health Professionals Communication Skills Scale(HP-CSS) [21,27] was self-administered. The 18-item HP-CSS instrument evaluates the communication skills that health professionals use to relate to their patients. The HP-CSS scale [27] was developed in a Spanish context and validated in a sample of health professionals (doctors, nurses, and nursing assistants), both in primary and specialized care. It is composed for four dimensions: the empathy dimension (score range, 5–30 points) explores how professionals obtain and provide information within the clinical relationship (items 2, 4, 6, 11, and 12); the informative communication dimension (6–36 points) evaluates the capacity to understand the feelings of patients and their behavioral responses of active listening and empathy (items 5, 8, 9, 14, 17, and 18); the respect dimension (3–18 points) addresses the respect shown by professionals in the clinical relationship (items 1, 3, and 15); and the social skill or assertiveness dimension (4–24 points) assesses social skills or the capacity for assertiveness in clinical relationships with patients (items 7, 10, 13, and 16); responses follow a 6-point Likert-type scale (1 = almost never to 6 = many times). Reported internal consistency values were 0.77 for the empathy dimension, 0.78 for informative communication, 0.74 for respect, and 0.65 for social skill or assertiveness. There is adequate evidence of its external validity, and all dimensions have demonstrated statistically significant correlations with the Maslach Burnout Inventory scale [28].

### 2.3. Procedure

The above set of questionnaires was administered by e-mail to the selected participants (using Google Form) along with a description of the study and an informed consent form, including a link for uploading the completed questionnaires and form. In order to encourage a high response rate, participants were sent three reminders (at one-week intervals) in an email that also contained the link for uploading the completed questionnaires.

All participants signed their informed consent to the study, which was conducted in accordance with the Helsinki Declaration and European Union Good Clinical Practice and was approved by the Bioethics Committee of the University of Alicante (UA-2018-10-24). The confidentiality of the data was guaranteed, and participants were assured that they could withdraw from the study at any time without prejudice.

### 2.4. Data Analysis

The free R program (version 3.4.0) [29] was used for confirmatory factorial and internal consistency analyses. We analysed the performance of the measurement instrument, calculating the data asymmetry, kurtosis, floor and ceiling effects. It can be considered normal distribution of variables when the coefficients of asymmetry and kurtosis are between −1.5 to 1.5 [30]. According to the literature [31], the floor or ceiling effects are defined as participant responses in excess of 15% in the lower or upper response category ranges, respectively, suggesting a reduced capacity to differentiate scores. Therefore, data were considered to be ordinal according to the criteria of Rhemtulla et al. [32] Estimates were obtained using the robust weighted least squares mean- and variance-adjusted (WLSMV), used with ordinal variables [32], from the Lavaan package in R. The fit of data to the model was analysed by using the comparative fit index (CFI), Tucker–Lewis Index (TLI), and root mean square error of approximation (RMSEA) [33], considering index values > 0.90 [34] and RMSEA values of 0.05–0.08 [33] to be adequate. The internal consistency was determined using the non-linear reliability estimator in the semTools package [35,36,37]. The ordinal alpha result, characteristic of tau-equivalent models, was also included for comparative purposes [37].

The variable attitude towards communication skills was selected to evaluate the construct validity of the instrument, using Pearson’s correlation coefficient (*r*).

Finally, the free R program was also used to calculate means and standard deviations for the scale and subscale items on the perception of communication skills in nursing students.

## 3. Results

### 3.1. Socio-Demographic Data

The study included 289 participants: 81.70% were female (*n* = 236), the mean age was 22.71 (*SD* = 4.87) years (range, 18–53 years), 96.54% (*n* = 279) had Spanish nationality, 94.14% were single, 5.20% were married or had a common-law partner, and 0.7% were separated or divorced. The average socioeconomic level was 8.16 (SD = 1.98; range = 0–13). As can be seen, this is a homogeneous sample. No significant differences were found between the different socio-demographic variables and the communication skills perceived by the students.

Almost all participants (91.7%) had received training in social skills during their degree course and 17% (*n* = 49) in another setting; only 7.17% (*n* = 21) had received no type of social skills education/training.

A mean score of 55.79 (SD = 2.84) points (range, 45–60 points) was obtained for the Attitudes towards communication skills scale.

### 3.2. Performance of the Scale and Psychometrics Proprieties

Table 1 lists the scale and subscale scores. Ceiling effects were observed in 15 of the 18 scale items.

All analyses were based on a second-order factorial model. We first tested the model proposed for the original structure without correlated errors in order to verify whether the modification indexes indicated covariance of errors between the same items. However, confirmatory factorial analysis of the questionnaire did not show an adequate fit of data to the congeneric measurement structure (chi-square = 461.644; df = 131; CFI = 0.91; TLI = 0.89; RMSEA = 0.09) (Table 2). Estimated factorial loads ranged from 0.11 to 0.87 (Figure 1), suggesting violation of the assumptions of a tau-equivalent model. After statistical verification, adequate indices were again not observed for this model (chi-square = 805.719; df = 146; CFI = 0.82; TLI = 0.81; RMSEA = 0.13). In the congeneric model, modification indices suggested the inclusion of covariance of errors between items 16 and 18, which produced the greatest change in chi-square value (*MI* = 241.031). The remaining modification indices suggested were <35,000. An adequate fit of data to the structure was observed for the model with the aforementioned correlated errors (items 16 and 18): chi-square = 220.613; df = 130; TLI = 0.98, CFI = 0.97, and RMSEA = 0.05 (95% CI = 0.04–0.06). Items 16 and 18 showed small factorial loads (0.11 and 0.12, respectively) with their dimensions (Figure 1).

The internal consistency according to the nonlinear SEM reliability coefficient was 0.83 for the empathy dimension, 0.64 for the informative communication dimension, 0.72 for the respect dimension, 0.62 for the social skills or assertiveness dimension, and 0.88 for the total scale, specifying the SEM model (Table 2).

Table 3 exhibits the correlations among the different scale dimensions, showing that all factors are inter-related with statistically significant moderate correlations. Table 3 displays the concurrent validity results, showing statistically significant correlations (range, 0.18 and 0.38) of the dimension scores and total score with the variable “attitude towards communication” (*r* = 0.35, *p* < 0.005).

### 3.3. Descriptive Statistics for Communication Skills 

Table 4 reports the mean values for the instrument dimensions. A high mean score of 4.99 (*SD* = 0.45) (out of maximum of 6) was obtained for the students’ perception of their communication skills in general and for the remaining dimensions, using the transformed score. The lowest mean score (*M* = 4.18; *SD* = 0.45), although still moderate, was obtained for the assertiveness dimension. 

## 4. Discussion

Confirmatory factorial analysis showed that the scale preserved the original structure proposed by Leal-Costa et al. [38], and the fit indices were adequate. The HP-CSS comprised the four dimensions identified in the original article: Informative communication, Empathy, Respect, and Assertiveness. However, fitting the data to the model required specification of a structure with correlated measures between items 16 (‘I find it difficult to make requests of patients’) and 18 (’I find it difficult to ask for information from the patients’). This change was suggested by the modification indices obtained after the confirmatory factor analysis and was previously reported in the study of health care professionals [21] as well as in the specific nursing professional sample [38]. In addition, it is important to note that both items had low factorial loads in their corresponding subscales and had also shown the lowest factorial loads in the other two validation studies. Nevertheless, item 18 was preserved because of its contribution to content validity [21]. Caution has been expressed about this covariance and the risk of compromising the factorial structure, given that each item belongs to a different subscale [39]. It has been proposed that the correlations result from their sharing of the same heading or that they belong to a common subscale [39]. At any rate, given that these correlations are observed in other samples [21,38], the rewording of items 16 and 18 is recommended in future studies [40]. Unlike the other validation studies in different samples, there was no need to include additional correlations to obtain an adequate fit, yielding a more parsimonious model with fewer model re-specifications.

The one dimensionality of the measurement instrument was analysed and confirmed, with the subscales proposed as second-order factors underlying a general factor that evaluates Communication skills. This new model has been examined in a final analysis of the measurement instrument in a sample of nursing professionals [38]. This is a major advantage for the implementation of the scale, offering a single total score for communication skills.

With respect to internal consistency, the fit of values was based on the specified SEM model, as recommended [41,42]. The results show a moderate–high score for the total scale and subscales *Empathy* and Respect. However, the internal consistency for the subscales Informative Communication and Social Skills did not reach values of 0.7, being acceptable when the value is ≥0.7 [43,44]. These consistency results are similar to those obtained for the original version [21], in which the Social Skills subscale was also below the value of 0.7. A lower value was observed for the Informative Communication subscale in comparison to the original version. Nevertheless, analysis of the internal consistency using the alpha coefficient showed that values were closer to those obtained by Leal-Costa et al. [21]. Utilization of the alpha coefficient when the tau-equivalence supposition is not met, specifically when there are correlated errors, has been reported to have severe effects on the estimation of internal consistency and may overestimate the reliability [45]. Items 18 and 16 may be responsible for the unacceptable internal consistency values obtained, and future studies should address the need for modifications to subscales on the ability to understand the feelings of patients (Informative Communication) and on the capacity for assertiveness and appropriate behaviours in the clinical relationship with patients (Social Skills) in order to achieve adequate internal consistency of ≥0.7 [43,44] in all subscales.

In the analysis of convergent validity, statistically significant positive correlations were found between scores for the Attitudes towards Medical Communication Scale [26] and the total score and subscale scores of the HP-CSS. Accordingly, a better attitude of the students towards communication skills was related to an improved perception that they possess them. In this regard, attitude is considered one of the predictive variables of behaviour according to the Theory of Planned Behavior [46,47,48]. These results have important implications in the educational setting, supporting proposals of the European Higher Education Area [15] on the importance of the attitudinal factor.

With respect to the general description of skills in nursing students, high scores were obtained for their perception of communication skills, similar to some previously published findings in nursing students [22,49]. However, a study of students in the final year of nursing and psychology courses reported low scores for social competence using a global index of social skills [18]. It is possible that the specific theoretical training of nursing students and their clinical practice sessions enhance their perception of communication skills for effective nurse–patient relationships. High scores were also obtained for the attitudinal factor, suggesting that the students consider communication skills to be relevant. Communication between health professionals and patients is one of the main health tools, as it directly affects the efficiency of health professionals’ interventions [27].

Comparable results were observed for all items in the health care professionals measured with the same scale [21], although the scores were slightly higher in the present study of students. Students may be less conscious of their difficulties in interacting with patients or may be influenced by their more recent training on effective approaches to therapeutic relationships. However, further research is required to evaluate whether the self-perception of communication skills transfers to the real-life clinical setting and whether students with a higher self-perception are in fact more competent in communicating with patients

In any case, it is relevant for university professors to evaluate students’ competence in skills as a key component of the learning process [50], allowing them to know the perception of their ability to manage adequate communication with users, as well as to identify those aspects and/or situations in communication in which they present greater difficulty. With this, teachers could reflect on and readjust the contents of the educational programmes, with repercussions on an improvement in future professional clinical practice.

One limitation was the utilization of an incidental sample, based solely on the willingness of students to participate in the study, and studies of representative and randomized samples of students are needed to verify these results. The study was limited by its relatively low response rate, mainly in the 2018/2019 groups, even though a standardized data collection procedure was followed. This participation rate could be explained by the time selected for the data collection (June–October), as in those months students were already completing their degree studies and did not access their university email accounts as frequently. In future studies, it may be advisable to provide incentives to increase this rate.

## 5. Conclusions

This is the first study to validate a scale that evaluates the communication skills of nursing students in Spain, including attitudes towards communication as an additional measure of convergent validity. HP-CSS offers university teachers a rapid and easily applied method to evaluate the acquisition by nursing students of their communication skills and relationship with patients. Also, the results show a high level of perceived communication skills in the last course of nursing degree.

## Figures and Tables

**Figure 1 ijerph-17-07565-f001:**
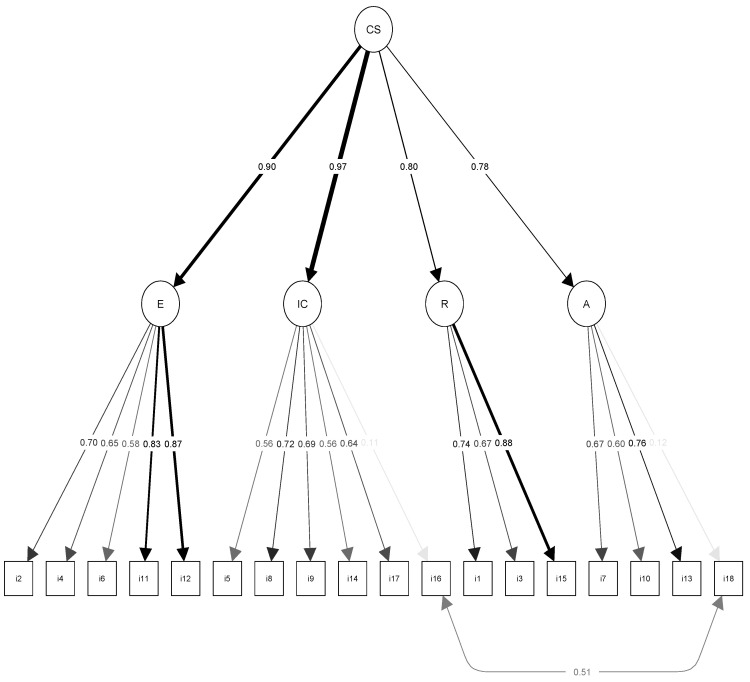
Confirmatory factor analysis of the original structure proposed by Leal et al. [22] graph extracted via the Lavaan package in the R freeware; CS = Health Professionals Communication Skills Scale E = Empathy; IC = Informative Communication; R = Respect; A = Social Skills or Assertiveness.

**Table 1 ijerph-17-07565-t001:** Performance of Health Professionals Communication Skills Scale (HP-CSS) and related normative data.

Item	Min	Max	*M* (*SD*)	Skewness	Kurtosis	Floor Effect *n* (%)	Ceiling Effect *n* (%)
Item 1	3	6	5.73 (0.50)	−1.84	3.52	1 (0.34)	220 (75.60)
Item 2	1	6	4.65 (1)	−0.52	0.33	2 (0.69)	62 (21.31)
Item 3	3	6	5.61 (0.61)	−1.38	1.23	1 (0.34)	195 (67.01)
Item 4	3	6	5.69 (0.52)	−1.52	2.25	1 (0.35)	207 (71.13)
Item 5	3	6	5.29 (0.72)	−0.66	−0.27	3 (1.03)	127 (43.64)
Item 6	2	6	5.40 (0.82)	−1.50	2.20	2 (0.69)	163 (56.01)
Item 7	1	6	4.24 (1.04)	−0.27	−0.15	2 (0.69)	32 (10.99)
Item 8	2	6	4.76 (0.96)	−0.56	−0.11	5 (1.72)	67 (23.02)
Item 9	3	6	5.44 (0.69)	−1.13	1.16	5 (1.72)	155 (53.26)
Item 10	1	6	3.74 (1.17)	−0.38	−0.30	12 (4.12)	12 (4.12)
Item 11	2	6	4.97 (0.98)	−0.79	0.14	5 (1.72)	101 (34.71)
Item 12	2	6	5.35 (0.80)	−1.23	1.52	5 (0.69)	151 (51.89)
Item 13	1	6	4.59 (0.99)	−0.47	−0.02	1 (0.34)	52 (17.89)
Item 14	4	6	5.87 (0.36	−2.57	5.91	2 (0.69)	254 (87.29)
Item 15	3	6	5.61 (0.57)	−1.26	1.18	1 (0.34)	190 (65.29)
Item 16	1	6	3.74 (1.36)	−0.10	−1.06	10 (3.44)	25 (8.59)
Item 17	2	6	5.14 (0.83)	−0.78	0.29	1 (0.34)	110 (37.80)
Item 18	1	6	4.15 (1.33)	−0.35	−0.99	3 (1.03)	45 (15.43)

Note: M = Mean; SD = Standard deviation; Min = Minimum; Max = Maximum.

**Table 2 ijerph-17-07565-t002:** Confirmatory analysis and internal reliability consistency (*n* = 289).

	Fitted Model	Chi-Square	df	CFI	TLI	RMSEA [95% CI]	Nonlinear Reliability ^a^	Ordinal Alpha ^a^
	TM	805.719	146	0.82	0.81	0.13 (0.12–0.13)	-	
	CM	461.644	131	0.91	0.89	0.09 (0.08–0.10)	-	
Total scale	CE	220.613	130	0.98	0.97	0.05 (0.04–0.06)	0.88 *	0.89 *
Empathy	CE	-	-	-	-	-	0.83	0.83 *
Informative communication	CE	-	-	-	-	-	0.64	0.71 *
Respect	CE	-	-	-	-	-	0.72	0.81 *
Social skill	CE	-	-	-	-	-	0.62	0.59 *

Note: TM = Essentially tau-equivalent measures; CM = Congeneric measures; CE = Measures with correlated errors (i16–i18); df = degrees of freedom; CFI = Comparative fix index; TLI = Tucker-Lewis index; RMSEA = Root mean square error of approximation; CI = Confident interval; ^a^ Reliability based on Structural Equation Model; * Ordinal Alpha added for comparison purposes.

**Table 3 ijerph-17-07565-t003:** Correlations between HP-CSS and attitudes to communication skills.

	E	IC	R	A	Total	Attitudes
E	1					
IC	0.55 **	1				
R	0.56 **	0.41 **	1			
A	0.44 **	0.59 **	0.29 **	1		
Total	0.82 **	0.84 **	0.63 **	0.79 **	1	
Attitudes	0.38 **	0.27 **	0.29 **	0.18 *	0.35 **	1

Note: E = Empathy; IC = Informative Communication; R = Respect; A = Social Skill/Assertiveness; Attitudes = Attitudes towards Communication Skills; * *p* < 0.005; ** *p* < 0.001.

**Table 4 ijerph-17-07565-t004:** Descriptive statistics of HP-CSS.

	Mean (SD)	Transformed Score	Range	P25	P75
Empathy	26.05 (3.01)	5.21 (0.6)	5–30	18	32
Informative communication	30.66 (2.86)	5.11 (0.48)	6–36	28	32
Respect	16.96 (1.31)	5.65 (0.44)	3–18	16	18
Social skill/Assertiveness	16.34 (3.01)	4.09 (0.73)	4–24	15	19
Total	90.00 (8.04)	5.00 (0.45)	18–108	85	96

Note: SD = Standard deviation; Transformed score (range 1–5) = Total scale score/number of items; p25 = 25th Percentile; p75 = 75th Percentile.

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
