# Peer review of "Psychometric Properties of the Health Professionals Communication Skills Scale in University Students of Health Sciences"

_ijerph, 2020, doi:10.3390/ijerph17207565_

Round 1
Reviewer 1 Report
Thank you for giving me an opportunity to review this manuscript.
This study has validated a health professionals communication skills scale (HP-CSS) among nursing students in Spanish. The author’s methodological process was almost appropriate, however, I have a few concerns below.
- Add an explanation of the difference between HP-CSS and Attitude Toward medical Communication Scale(ref.no 25). What is a need to use HP-CSS?
- In a study of No.21(Leal-Costa, et al 2016), they had already validated HP-CSS among nurses and nursing assistants. What is the need the authors conducted this study? Are nursing students different from nurses?
- I have a concern about the low response rates in this study. What kind of bias would you infer? Add it in limitation.
- In line 86, page 2, check a grammatical error: “with ~ being is”.
- In line 108, add the reference paper to Maslash Burnout Inventry Scale.
- In line 131 and 148, “attitude towards communication skill”, Should the first letter be uppercase?
- Foot note in Table 1, is the ref no.27 right?
Author Response
Please see the attaachment

Reviewer 2 Report
Review of the manuscript:
Psychometric properties of the health professionals' communication skills scale in university students of health sciences
Summary:
This study evaluates the dimensions of communication skills perceived by health professionals employing different questionnaires. Also, it emphasizes the importance of non-technical training, such as interpersonal skills. The volunteers of this study were students in their fourth year of nursing at two Spanish universities. However, the results are not well presented and well-arranged, thus, not clear enough to the reader.
Numerous statistical statements and reports have been used without indicating how they are interpreted and what they are implying for, which might not be clear to the readers. This study is based on the nursing students' judgment about their communication skills and nurse-patient relationship, including respect and empathy. However, one cannot be conscious enough of their behavior towards patients, and such evaluations could be best performed by the patients.
Page 3- L96: Communications skills should be changed to communication skills.
Page 4- L148: How would the students' nationality influence the final analysis while almost all of the volunteers are from the same nationality (the same question applies to marital status). Unnecessary data is shown in the table.
Page 5- L 168: There should be more clarification on the self-administered 18-item HP-CSS, and how the data could be interpreted from Table 2. The table is not self-explanatory, and also the caption does not provide sufficient information on the normative data and the correlation between the values.
Page 5- L171: The caption used for Figure 1 is not informative enough, as the figure is not self-explanatory; Which is typically expected from this figure, helping readers understand the methodology better.
Page7- L226: Please explain what the 0.7 value indicates. It should be clear what the range below or above 0.7 implies.
The results could also be presented figuratively by utilizing students' scores in dimensions such as informative communication, empathy dimension. To better visualize students' competency in each dimension of communication skills.
As mentioned in the methods, The Family Affluence Scale (FAS_III) has been used to assess family socioeconomic status. However, the probable effects of the family's economic well-being in the students' competence in an effective nurse-patient relationship are not mentioned in the results.
Author Response
|
Reviewer 2 |
|
|
The results are not well presented and well-arranged, thus, not clear enough to the reader.Numerous statistical statements and reports have been used without indicating how they are interpreted and what they are implying for, which might not be clear to the readers. |
We have tried to explain better the interpretation of the data in data analysis. Line 129-135 |
|
This study is based on the nursing students' judgment about their communication skills and nurse-patient relationship, including respect and empathy. However, one cannot be conscious enough of their behavior towards patients, and such evaluations could be best performed by the patients. |
Line 278-283: A sentence has been included at the end of the discussion to justify how it might impact on improvement with patients: “In any case, it is relevant for teachers to evaluate students' competence in skills as a key component of the learning process (49), allowing them to know the perception of their ability to manage adequate communication with users, as well as to identify those aspects and/or situations in communication in which they present greater difficulty. With this, teachers could reflect on and readjust the contents of the educational programmes, with repercussions on an improvement in future professional clinical practice”. |
|
Page 3- L96: Communications skills should be changed to communication skills. |
Amended, line 101 |
|
Page 4- L148: How would the students' nationality influence the final analysis while almost all of the volunteers are from the same nationality (the same question applies to marital status). Unnecessary data is shown in the table. |
The authors have presented the socio-demographic data to show the reader the characteristics of the participants, noting that it is a sample with homogeneous characteristics. This information has been included in the text, removing it from the table. Furthermore, it has been specified that there is no statistically significant correlation between any of the sociodemographic variables regarding the communication skills of the students. We have included a sentence to describe in general way for the socio-demographic variables, line 153: “As can be seen, this is a homogeneous sample. No significant differences were found between the different socio-demographic variables and the communication skills perceived by the students”. |
|
Page 5- L 168: There should be more clarification on the self-administered 18-item HP-CSS, and how the data could be interpreted from Table 2. The table 2 is not self-explanatory, and the caption does not provide enough information on the normative data and the correlation between the values. |
We have expanded the information on the scale to facilitate potential readers understand the construct and scale aim. Lines 101 – 105: “The Communication Skills Scale in Health Professionals (HP-CSS) [21, 27] was self-administered. The 18-item HP-CSS instrument evaluates the communication skills that health professionals use to relate to their patients. The HP-CSS scale [27] was developed in a Spanish context and validated in a sample of health professionals (doctors, nurses, and nursing assistants), both in primary and specialized care. It is composed for four dimensions: …” Also, we have added information in the data analysis section (lines 130-135) to facilitate the interpretation of table 2. |
|
Page 5- L171: The caption used for Figure 1 is not informative enough, as the figure is not self-explanatory; Which is typically expected from this figure, helping readers understand the methodology better. |
We consider that the description of the figure should contain only the title and the acronym for its understanding. Figure 1 and the results are already described in the text. We suggest that, to improve the readers' understanding, it would be convenient for table 1 to be just after the first paragraph of the results (which are only two lines) and after the second paragraph of the results (where figure 1 is explained) to insert figure 1. |
|
Page7- L226: Please explain what the 0.7 value indicates. It should be clear what the range below or above 0.7 implies. |
It has been included in line 241 that the acceptable value for internal consistency is ≥ 0.7. |
|
The results could also be presented figuratively by utilizing students' scores in dimensions such as informative communication, empathy dimension. To better visualize students' competency in each dimension of communication skills. |
We appreciate the suggestion. However, in table four the data are already presented by dimensions, so we consider that the presentation in a figure would mean repeating information. If the editor feels that it provides new information, we would include it. |
|
As mentioned in the methods, The Family Affluence Scale (FAS_III) has been used to assess family socioeconomic status. However, the probable effects of the family's economic well-being in the students' competence in an effective nurse-patient relationship are not mentioned in the results. |
The authors have presented the sociodemographic data to show the reader the characteristics of the participants, observing that it is a sample with homogeneous characteristics. In any case, the variables socioeconomic level and competence perceived in communication skills by the students are not correlated (r= 0.01). We have included a sentence to describe in general way for the socio-demographic variables, line 153: “As can be seen, this is a homogeneous sample. No significant differences were found between the different socio-demographic variables and the communication skills perceived by the students”.
|

Reviewer 3 Report
Presented paper describes emerging problem of healthcare paradigm where clinical skills prioritize over non-technical (communication) skills. Raising proper questions is a good step in changing the model.
With undoubted importance of the work, some minor issues should be clarified:
L67 Why fourth year nursing were chosen? If that is the last study year in announced universities - please state that clearly for different programs have different timespan.
L86 Why such an unusual scoring (0-13) If that is a maximum sum of "well-being items" please explain extra point: so far it looks like maximum sum =12.
Best regards.
Author Response
|
Reviewer 3 |
|
|
L67 Why fourth year nursing were chosen? If that is the last study year in announced universities - please state that clearly for different programs have different timespan. |
Page 2, line 68: “All students in their fourth year of nursing at two Spanish universities were invited to participate. In the Spanish university environment, the fourth year is the last course of the nursing degree studies. Fourth-year students were chosen to ensure that, in spite of possible differences in their curricula, both universities had addressed content related to communication skills in different situations, as well as having carried out more than 75% of their clinical practice.” |
|
L86 Why such an unusual scoring (0-13) If that is a maximum sum of "well-being items" please explain extra point: so far it looks like maximum sum =12. |
Line 88: It was a typographical error. Is has been corrected. The variable "number of computers" has four response options (0,1,2,3). Therefore, the sum of all the items is 13. |